# Effect of Cellulose Nanofibrils and TEMPO-mediated Oxidized Cellulose Nanofibrils on the Physical and Mechanical Properties of Poly(vinylidene fluoride)/Cellulose Nanofibril Composites

**DOI:** 10.3390/polym11071091

**Published:** 2019-06-27

**Authors:** Eftihia Barnes, Jennifer A. Jefcoat, Erik M. Alberts, Mason A. McKechnie, Hannah R. Peel, J. Paige Buchanan, Charles A. Weiss Jr., Kyle L. Klaus, L. Christopher Mimun, Christopher M. Warner

**Affiliations:** 1Geotechnical and Structures Laboratory, U.S. Army Engineer Research and Development Center, 3909 Halls Ferry Road, Vicksburg, MS, USA; 2HX5, LLC, Vicksburg, MS, USA; 3Environmental Laboratory, U.S. Army Engineer Research and Development Center, 3909 Halls Ferry Road, Vicksburg, MS, USA

**Keywords:** cellulose nanofibrils, poly(vinylidene fluoride), solvent evaporation, surface wettability, mechanical properties

## Abstract

Cellulose nanofibrils (CNFs) are high aspect ratio, natural nanomaterials with high mechanical strength-to-weight ratio and promising reinforcing dopants in polymer nanocomposites. In this study, we used CNFs and oxidized CNFs (TOCNFs), prepared by a 2,2,6,6-tetramethylpiperidine-1-oxyl radical (TEMPO)-mediated oxidation process, as reinforcing agents in poly(vinylidene fluoride) (PVDF). Using high-shear mixing and doctor blade casting, we prepared free-standing composite films loaded with up to 5 wt % cellulose nanofibrils. For our processing conditions, all CNF/PVDF and TOCNF/PVDF films remain in the same crystalline phase as neat PVDF. In the as-prepared composites, the addition of CNFs on average increases crystallinity, whereas TOCNFs reduces it. Further, addition of CNFs and TOCNFs influences properties such as surface wettability, as well as thermal and mechanical behaviors of the composites. When compared to neat PVDF, the thermal stability of the composites is reduced. With regards to bulk mechanical properties, addition of CNFs or TOCNFs, generally reduces the tensile properties of the composites. However, a small increase (~18%) in the tensile modulus was observed for the 1 wt % TOCNF/PVDF composite. Surface mechanical properties, obtained from nanoindentation, show that the composites have enhanced performance. For the 5 wt % CNF/PVDF composite, the reduced modulus and hardness increased by ~52% and ~22%, whereas for the 3 wt % TOCNF/PVDF sample, the increase was ~23% and ~25% respectively.

## 1. Introduction

The behavior of composite materials is strongly influenced by the physical and chemical properties of its constituents as well as the interfacial interactions between them. Incorporating dopants in a polymeric matrix can lead to significant changes in mechanical strength, toughness, thermal or electrical conductivity, optical transparency, dielectric properties, etc. [1,2,3,4,5,6] Lightweight fillers, acting as reinforcing agents in polymeric matrices, are of particular interest in many research fields, since they can improve physical and mechanical properties of the composites without substantial weight increase [1]. Reinforcing fillers include clays [2,3], carbon nanomaterials such as carbon nanofibers [4] and carbon nanotubes [5], which have been widely studied in various polymer systems and composites [6]. However, some of these fillers are recently discovered to be toxic and prone to bioaccumulation [7]. For example, the National Institute of Occupational Safety and Health (NIOSH) has made recommendations for reducing worker exposure to carbon nanotubes and nanofibers, as well as respiratory screening for workers exposed to 1 μg/m^3^ (8 hour time weighted average) [8,9]. Certain nanoclays, such as Cloisite Na^+^, Cloisite 93A and Cloisite 30B, have been reported to induce significant cell death, genotoxic effects and morphological alterations in human cell lines [10]. Hence, in recent years, there has been a growing interest in abundant and environmentally friendly materials that can be used as additives in functional polymers. One of these additives is cellulose, which is the most abundant organic polymer on Earth and is the basic structural building block of plants and biofilms [11]. Cellulose consists of anhydroglucose ring units that assemble into nanofibrils, consisting of several amorphous and crystalline domains [11]. Acid hydrolysis breaks down the amorphous domains, leaving behind crystalline, rod shaped particles known as cellulose nanocrystals (CNCs). CNCs are lightweight, have a large surface area and high tensile strength and Young’s modulus; all these properties make them promising reinforcing nanofillers [12]. Conversely, high-aspect ratio cellulose nanofibrils (CNFs) exhibit reinforcing effects that have been attributed to the presence of entangled CNF networks capable of bridging crazes formed in composites during tensile testing [13]. The high-aspect ratio of CNFs facilitates fibril interlocking across crazes as well as forming entangled fibril pullouts that can contribute to the increased tensile strength and strain-at-failure at low loadings [13,14]. 

Poly(vinylidene fluoride), or PVDF, is a technologically important polymer with outstanding electrical and chemical resistance properties [15]. PVDF possesses ferroelectric, pyroelectric and piezoelectric properties and has been used as the building block of smart materials such as actuators [16,17], transducers [18], sensors [19,20], nonvolatile memory [21,22], energy harvesting and storage applications [23,24,25,26]. PVDF is extensively used as a membrane or scaffold for water purification and filtration technologies. PVDF is a semi-crystalline polymer with five known crystalline phases (α-, β-, γ-, δ- and ε-) whose nucleation is highly dependent on the processing conditions [14]. The most common phases are the non-polar, non-electroactive α-phase with antiparallel TGTG′ (trans-gauche–trans-gauche) chain formation, the electroactive β- and γ-phases with trans TTT planar and T_3_GT_3_G′ respective chain conformations [14,27,28]. Notably, transformation between various PVDF phases can be achieved by incorporation of additives such as hygroscopic salts or charged nanoparticles [14].

Recent studies on cellulose/PVDF composites have demonstrated that nanocellulose enhances nucleation of polar crystalline phases; this enhancement was attributed to its large surface area and availability of the surface hydroxyl groups [29,30]. CNCs have been shown to enhance crystallinity, mechanical and thermal stability in CNC/PVDF composite films [31]. Addition of CNCs in electrospun PVDF increased both the relative fraction of β-phase and crystallinity of the composite fibers over those of neat PVDF fibers [32]. These results suggest that nanocellulose interacts with the PVDF matrix to enhance or modify some of its physical properties. Despite the similar cost of commercially available CNCs and CNFs (see Appendix A), the use of CNFs as a dopant or reinforcing filler in PVDF is limited [33]. Improvement of mechanical properties has been reported for PVDF composites with CNCs [31], carbon nanofibers [34], carbon fibers [35], graphene [36], and graphene oxide-titania nanolayers [37]. When PVDF was doped with carbon nanofibers, 122% increase in yield strength and 88% increase in tensile modulus was observed for 4 wt % carbon nanofiber loading [34]. Graphene oxide-titania PVDF composites exhibited ~115% increase in tensile strength and ~270% increase in tensile modulus [37]. Incorporation of dopants such as carbon fibers, graphene and CNCs in PVDF, resulted in a modest increase of tensile properties of the composites (between 10% and 20%) [31,35,37]. The nanoscale mechanical properties of PVDF/MWCNT composites were probed with atomic force microscopy-based nanoindentation, with the composite films showing enhancement of both hardness and modulus over that of neat PVDF [38]. 

The goal of this study is to investigate the effect of CNFs on the physical properties of PVDF composites. CNFs with neutral surface chemistry and TOCNFs with negative surface chemistry were dispersed in PVDF dissolved in *N*,*N*-dimethyl formamide (DMF), using a high-shear mixing method, followed by doctor blade casting and drying, to obtain composites films with up to 5 wt % cellulose nanofibrils. The negatively charged TOCNFs are hypothesized to strongly interact with the positive CH_2_ PVDF dipoles and affect the crystalline structure of the composites. We examined the surface morphology of the films as well as physical properties such as the crystalline structure, surface wettability, crystallinity, thermal and mechanical behaviors as a function of cellulose loading and cellulose surface chemistry/treatment. Distinct morphologies, thermal and mechanical behaviors were observed for the CNF/PVDF and TOCNF/PVDF composites.

## 2. Materials and Methods 

### 2.1. Materials

PVDF pellets with an average molecular weight of ~180,000 g/mol (*M*_w_) and 71,000 g/mol (*M*_n_) were purchased from Sigma Aldrich (St. Louis, MO, USA). Wood pulp CNF slurry (~4 wt % solids in water) was obtained from the University of Maine Process Development Center (Orono, ME, USA) and stored in a refrigerator at 4 °C until use. DMF (anhydrous grade, 99.8%) was purchased from Sigma Aldrich (St. Louis, MO, USA) and used immediately after opening. Through six cycles of DMF addition, centrifugation, and decantation, the aqueous CNF slurry was solvent exchanged in DMF, prior to incorporation into the PVDF/DMF solution. TOCNFs were prepared by mixing 2,2,6,6-tetramethylpiperidine-1-oxyl radical (TEMPO, 0.06 g) sodium bromide solids (NaBr, 0.3 g) and bleach (6 wt % NaClO, 37.22 g,) to the CNF slurry (3 g) under vigorous stirring. After one hour, 10 mL of 0.5 M NaOH solution were added, followed by another 10 mL of the 0.5 M NaOH solution at two hours. After five hours, 15 mL of ethanol were added to quench the reaction. The TOCNFs were centrifuged and rinsed with deionized water (18.2 MΩ) three times before exchanging the water with DMF. The TOCNF slurry had a milky white color similar to the color of the as-received CNFs. 

### 2.2. Material and Film Preparation 

A 10 wt % PVDF/DMF solution was obtained by mixing the solvent and the polymer pellets at 60 °C for 8 h. CNF/PVDF and TOCNF/PVDF composite films were obtained by adding a known amount of cellulose into the PVDF/DMF solution. The wt % reported here is that of the cellulose solids relative to the total weight of the composite. The solutions were mixed at 2500 RPM for 5 mins and were then cast onto clean glass plates using a #30 stainless steel casting blade (Gardo). Solvent evaporation was achieved at 100 °C overnight. The plated dry films were soaked in methanol for 30 min to aid in delamination, and free-standing films, with approximate thickness between 50–60 µm, were obtained after residual methanol removal overnight at 60 °C. 

### 2.3. Moprhological Characterization 

Low magnification optical images of the dried CNF slurry were collected with a polarizing optical microscope (Imager.Z1m, Zeiss, Gottingen, Germany). Briefly, dilute CNF solution was dispersed on a glass slide and left to dry before collecting darkfield images under 20× and 50× optical magnification. The morphologies of the CNFs and the films were characterized with a field emission scanning electron microscopy (Nova Nano-SEM 630, FEI) under the secondary electron (SE) imaging mode. The accelerating voltage was set between 5 kV and 10 kV and the spot size between 2.0 and 5.0. In order to reduce surface charging, the samples were coated with a thin layer of gold prior to SEM imaging. Higher resolution images of CNFs, TOCNFs and composite films were obtained with Atomic Force Microscopy (Dimension Icon, Bruker, Goleta, CA, USA). The CNFs and TOCNFs were imaged following the methodology outlined of Sacui et al. [39]. Briefly, poly-*L*-lysine (PLL) solution was drop-cast on freshly cleaved mica substrates, and after 5 min the substrates were dried with nitrogen gas. Dilute nanocellulose suspensions (~0.02–0.03 wt %) were drop-cast onto the PLL-coated mica substrates. After 5 min, the substrates were gently rinsed with DI to remove unattached cellulose and then blown dry. Topographical images were captured in the ScanAsyst® mode using a tip with a 2 nm nominal tip radius. The AFM data were processed and plotted with Gwyddion [40].

### 2.4. Contact Angle 

To evaluate the influence of the CNFs/TOCNFs on the wetting properties of the films, static water contact angle measurements were carried out with a drop shape analyzer (DSA) (DSA30S, Kruss). The contact angle was recorded immediately after a 5 µL water sessile droplet was gently dropped on the surface of the film. The reported contact angle values are the average of four measurements carried out on various locations on the films under ambient conditions. Contact angle measurements were carried out on the top (free) and bottom (previously constrained) surface of the free-standing films. Time-dependent contact angle measurements were carried out up to 120 s with a 1 s sampling rate. The contact angle time dependence was fitted using the following equation:(1)$$\theta \left(t\right)=\theta_{i}\left(t\right)e^{-kt^{n}}$$
where *θ_i_* is the initial water contact angle. *k* and *n* are the kinetic constant and the exponential parameter, respectively [41,42]. The contact angle *θ* decreases over time due to spreading, adsorption and evaporation. For the fittings, the value of *n* was constrained between 0 (pure adsorption) and 1 (pure spreading) [42].

### 2.5. Crystalline Structure

Information regarding the crystalline phase of the CNFs, before and after oxidation, as well as the neat and composite films was obtained with X-ray diffraction using a materials research diffractometer (Panalytical X’Pert Pro) equipped with a Co-K_α_ X-ray source operated at 45 kV and 40 mA. Diffraction patterns were obtained from 5 to 70° 2θ with a step size of 0.02°. Circular specimens were affixed to zero background holders made from 9N semiconductor grade silicon. The Co- K_α_ 2θ values were converted to the corresponding Cu-K_α_ values that are more commonly referenced in literature. The diffraction patterns have been vertically shifted for clarity. The cellulose crystallinity index (*CI*) was estimated with the Segal method [43]:(2)$$\mathrm{CI}=\frac{I_{002}{\;-\;I}_{amorphous}}{I_{002}}$$
where *I*_002_ is the maximum intensity of the (002) diffraction peak and *I*_amorphous_ is the intensity of the amorphous diffraction, which is taken at the 2*θ* angle between the (002) and (101) peaks with minimum intensity. Fourier transform infrared (FTIR) spectroscopy (Nicolet 6700 FTIR) was also carried out to estimate the fraction of the dominant crystalline phases present in the films as well as to probe specific dopant interactions with the PVDF matrix. Attenuated total reflectance FTIR (ATR-FTIR) spectra were collected from a sampling area of ~1.5 mm in diameter, with three spectra collected per sample. Since the intensity of the band at ~1070 cm^−1^ has a linear dependence on film thickness and is independent of the crystalline phases present, the spectra of all films were normalized with respect to the amplitude of that peak [15]. The crystalline fraction of the γ-phase (F(γ)) present in the films can be calculated from:(3)$$F\left(\gamma \right)=\frac{A_{\gamma }}{\left(K_{\gamma }{/K}_{\alpha }\right)A_{\alpha }{+A}_{\gamma }}$$
where *A*_γ_/*A*_α_ are the FTIR absorbance at 833 and 764 cm^−1^, respectively, and K_γ_/K_α_ are the absorption coefficients at 833 and 764 cm^−1^ [15].

### 2.6. Thermal and Mechanical Behavior

The melting behavior of the films was examined with differential scanning calorimetry (DSC250, TA Instruments; New Castle, DE, USA). A heat-cool-heat cycle was employed from -25 to 185 °C at 5°C/min heating and cooling rates. The peak maximum of the melting endotherm from the first heating segment was recorded and referred to as the *T*_m_, whereas the maximum of the crystallization endotherm is referred to as *T*_cryst_. The crystallinity (*X*_c_) was calculated from:(4)$$X_{c}=\frac{{\Delta H}_{m}/\phi }{{\Delta H}_{m}*}\times 100\%$$
where *ΔH*_m_ is the melting enthalpy measured from the differential scanning calorimetry curves, *ΔH*_m_* is the melting enthalpy for 100% polymer crystallinity, and *φ* is the weight fraction of PVDF in the films. From literature, *ΔH*_m_* is reported to be 104.5 J/g for the 100 % crystalline material [44].

The thermal stability of CNFs, as-received and oxidized, and that of the films was monitored with a thermogravimetric analyzer (STA449 F1 Jupiter, Netzsch) at a constant heating rate of 10 °C/min under 20 mL/min nitrogen. *T*_10%_ and *T*_50%_ are defined as the temperatures corresponding to 10 wt %, and 50 wt % weight loss, whereas *T*_p1_, *T*_p2_, etc. are the temperatures corresponding to the first, second, etc. peak of the DTG curve. Mechanical (stress versus strain) properties were evaluated in tension mode with a hybrid rheometer (TA Instruments Discovery Hybrid Rheometer) functioning as a dynamic mechanical analyzer in tension mode. Samples were cut into ~5 mm × 15 mm strips along the draw direction, and tests were conducted at a constant linear rate of 10 µm/s. The thickness of the specimens was measured prior to each of the tensile tests. The reported values are the average of three runs per film. The yield stress was calculated using the offset method at 0.2% strain, and the tensile toughness was calculated by integration of the area under the stress-strain curves. Surface mechanical properties (reduced modulus *E_r_* and hardness *H*) were determined with an Agilent G200 nanoindenter equipped with a pyramid-shaped diamond Berkovich indenter. Films were attached with epoxy on metal discs and load-controlled indentation measurements were performed with a maximum load of 10 mN, 20 s loading time, followed by a hold time of 10 s, and 10 s unloading time. A hold segment in air corrected for thermal drift by waiting until the thermal drift was less than 0.05 nm-s^−1^ before testing commenced. The reported *E_r_* and *H* values are averages over 25 indents obtained from a 5 × 5 indentation grid with 30 µm spacing between the indents. To minimize surface morphology effects on the force-displacement curves, nanoindentation was carried out on the smooth (bottom) surface of the composites.

## 3. Results and Discussion

### 3.1. Surface Characterization and Morphology

Figure 1a,b show the CNF slurry consists of a dense network of high aspect ratio CNFs. Not all fibrils appear to have nanoscale (<100 nm) dimensions, but the physical dimensions of CNFs can vary greatly depending on the degree of fibrillation, pretreatments and agglomeration [11]. Higher resolution AFM images reveal the presence of smaller cellulose particles whose morphology is consistent with that of CNCs (see Appendix A). The X-ray diffraction patterns of the CNFs and TOCNFs are shown in Figure 1c. The CNF diffraction pattern has a broad diffraction peak centered at ~15.8° as well as peaks at 22.7° and 34.6°. The broad peak at 15.8 °C is a convolution of two crystalline peaks corresponding to the cellulose I (101) and (10-1) planes, whereas the peaks at 22.7° and 34.6° correspond to the cellulose I (002) and (040) planes respectively [45]. The diffraction pattern of the TOCNFs shows an additional peak at 7.7°, with shoulders at 20.2° and 25.2°, as well as a shift of the highest intensity peak to 22.6°. It has been reported that alkali and acid treatments transform cellulose from type I to type II [46]; hence, the observed changes in the TOCNF diffraction pattern are indicative of cellulose II formation as well as an increase in amorphous content. The CI of the CNFs and TOCNFs was calculated to be 62.3% and 41.2%, respectively. In addition to the reduction of crystallinity, the oxidation treatment yields finer cellulose fibrils when compared to the CNFs (see Appendix A).

The FTIR absorbance spectra of dried CNFs and TOCNFs are shown in Figure 1d. The spectra exhibit characteristic cellulose absorption bands such as the −CH stretching peak at 2900 cm^−1^ and the −CH_2_ wagging peak at 1315 cm^−1^, as well as the broad peak between 3000 and 3650 cm^−1^ due to –OH stretching [47]. For the CNFs, the broad peak around 1645 cm^−1^ corresponds to the −OH bending mode of adsorbed water [48]. The spectrum of TOCNFs exhibits peaks at 1602 cm^−1^ and 1409 cm^−1^ corresponding to vibrations of surface carboxylate groups (–COONa), which verifies the oxidized state of the TOCNFs [49]. Unlike CNFs, which have neutral surface charged, TOCNFs have negative surface charges which can interact with the positive CH_2_ PVDF dipoles and induce crystalline phase changes in the composites.

Figure 2 and Figure 3 show photographs and SEM images of the neat PVDF and composite CNF/PVDF and TOCNF/PVDF films. The films are opaque with gradual color change towards pale yellow with increasing cellulose content. The morphology of the top surface and cross-sectional surface of the neat PVDF film is shown in Figure 2b,c. Higher magnification imaging of the top surface (inset of Figure 2b) reveals roughly spherically shaped, interconnected spherulites, which are typical of γ- PVDF [50]. The surface porosity of the composites increases with addition of CNFs, with some CNFs becoming visible on the surface (for example see Figure 2g,h). As shown in Figure 2i, the cross-sectional surface of the 5 wt % film has high aspect ratio fibrils, which are likely CNFs. The increase of surface porosity with a higher CNF content, may be attributed to CNF-bound water present in the films during solvent evaporation. Vapor-induced phase separation in PVDF occurs in the presence water, because the latter is miscible with hygroscopic DMF, but is a non-solvent for PVDF [51]. CNFs are hygroscopic and likely retain some water after solvent exchange and/or absorb water from the environment during solvent evaporation.

The morphology of the top surface of the TOCNF/PVDF films is shown in Figure 3. SEM of the cross-sectional surface of the 5 wt % TOCNF/PVDF film is shown in Figure 3g, revealing thin, high aspect ratio fibrils, which are likely TOCNFs. Under low TOCNF wt % (Figure 3b,c), the films exhibit disconnected regions with increased porosity. Similar to the CNF/PVDF films, the surface porosity is likely associated with increased water retention enabled by the presence of TOCNFs. As the TOCNF concentration increases, the top surface exhibits macroscopic disorder in the form of undulating peaks and valleys. TEMPO-mediated oxidized cellulose with –COONa surface groups has been reported to agglomerate in DMF, and it is likely the observed surface disorder is associated with TOCNF agglomeration [52]. In addition, TOCNF–COONa retains more water than other types of nanocellulose [53]. High resolution AFM images (see Appendix A) also capture the local morphology and surface roughness of the PVDF, CNF/PVDF, and TOCNF/PVDF top surfaces. 

The surface wettability of neat PVDF, CNF/PVDF, and TOCNF/PVDF films is shown in Figure 4. Neat PVDF exhibits a contact angle of 90.3° ± 3.9° on the top (Figure 4a) surface and 84.4° ± 0.7° on the bottom surface (Figure 4c), in good agreement with previous studies [31,54]. The contact angle difference between the top and bottom surface is due to the increased surface roughness of the top (free) surface, due to unconstrained PVDF crystallization as well as vapor-induced phase separation. The contact angle of the top surface of the CNF/PVDF composites stays approximately constant for the 0.5 wt % film and gradual increases with increasing CNF wt %, to a maximum contact angle of 103.5° ± 1.7° at 2 wt % (Figure 4b). Even though CNFs are hydrophilic, the local surface roughness increases with higher CNF content, resulting in an increase of the contact angle. This observation agrees with the Cassie-Baxter model, which predicts that for a rough hydrophobic surface a nonwetting liquid will not occupy surface cavities, resulting in increased hydrophobicity with increasing surface roughness [55]. 

On the contrary, the wetting properties of the bottom (constrained) surface show no appreciable trend with regard to the CNF wt %. For the TOCNF/PVDF composites, the top surface has a minimum contact angle of 85.5° ± 0.8° at 0.5 wt %, and a maximum value of 94.2° ± 1.5° at 2 wt %, and in general the contact angle does not show a strong correlation with the amount of added TOCNF. Further, the contact angles of the TOCNF/PVDF top surfaces are generally smaller than the contact angles of the CNF/PVDF top surfaces of the same wt %. Unlike the wetting properties of the bottom surface of the CNF/PVDF composites, the hydrophilicity of the TOCNF/PVDF bottom surface gradually increases, with increasing TOCNF wt % suggesting that TOCNFs are likely dispersed on the surface, but do not disturb its surface roughness. For more than 1 wt % loading, the contact angles of the TOCNF/PVDF bottom surfaces tend to be smaller than those of the CNF/PVDF bottom surfaces. A representative photograph of the measured contact angle for the 5 wt % TOCNF/PVDF is shown in Figure 4d. 

The time-dependent contact angle evolution of the top (free) and bottom (constrained) surfaces of PVDF, CNF/PVDF and TOCNF/PVDF films is shown in Figure 5. The fits, obtained from Equation (1), are shown as solid lines. Table 1 summarizes θ_i_, n and k (Appendix A includes error bars, adjusted regression coefficient r^2^ and standard error for each fit). For the bottom surface of the composites, the temporal evolution of the contact angle for all films (PVDF, CNF/PVDF, TOCNF/PVDF) is generally dominated by spreading (n ranges ~ 0.9–1.0) and small changes in the kinetic constant. On the contrary, the response of the top surface of the films is more complex. For some of the films, Equation (1) did not yield satisfactory fittings for a number of reasons. First, the model could not account for the observed rate of change in the contact angle during the first 20 s of recording (for example, see 3 wt % and 5 wt % sample response in Figure 5a). Second, for some of the films, air bubbles began to form inside the droplet, indicating that the droplet was gradually wetting internal cavities. This indicates a transition from an initial Casie-Baxter state to Wenzel state. Formation of bubbles was typically followed by discrete depinning and spreading of the droplet due to water filling cavities in the composites. These depinning events are shown as discrete changes in the measured contact angle. For example, in Figure 5b, the 4 wt % sample undergoes at lest two depinning events over 120 sec. For the time-dependent contact angle behavior succefully fitted with Equation (1), addition of hydrophilic CNFs or TOCNFS, resulted in a mixed regime of droplet adsorption and spreading on the top surface of the composites, and increase in the kinetic constant. Further, the kinetic constant of the top surfaces was larger than that of the bottom surfaces.

### 3.2. Crystalline Structure 

X-ray diffraction patterns obtained from the neat and composite films are shown in Figure 6. In general, all of them show two prominent peaks at approximately 18.7° and 20.3°, which correspond to the (020) and (110) reflections of γ-PVDF [56]. The smaller peak at ~39° is also assigned to the (210) γ reflection [15,56]. As the cellulose wt % increases, a small shoulder on the higher 2θ side of the main PVDF peak emerges and corresponds to the (020) cellulose reflection. For our processing conditions, all the films, regardless of the amount of added cellulose, remain in a majority γ-phase. Conversion of γ-phase PVDF into another crystalline phase, such as β-phase, can be achieved with the addition of hygroscopic dopants or negatively charged particles [57,58]. For example, addition of hygroscopic calcium chloride in PVDF/DMF/acetone solutions led to the formation of high content β-phase composites [57]. It is hypothesized that hygroscopic salts retain water during the drying process, which enable hydrogen bonding between the water molecules and the CF_2_ dipoles of PVDF [57]. To date, no report has been published on hygroscopic cellulose inducing polar to polar (γ- to β-) phase transformation in PVDF. CNC/PVDF and CNC/PVDF/PMMA blends remain in the same phase as that of neat PVDF [31,32,59]. Addition of non-nanoscale cellulose into electrospun PVDF did not affect either the crystalline phase or the crystallinity of the composite fibers [60]. 

The surface charge of nanoparticles plays an important role on PVDF phase transformations. It has been reported that negatively charged nanoparticles induced γ-phase transformation under processing conditions that normally crystalize α-phase PVDF, whereas positively charged nanoparticles do not induce a phase transformation [15]. The proposed mechanism responsible for the phase transformation is electrostatic interactions between negatively charged nanoparticles (ions) and positively charged CH_2_ dipoles [61]. Nanocellulose was recently shown to promote polar phase formation when addded to α-phase PVDF [29,30].

The FTIR spectra of neat PVDF and CNF/PVDF films are shown in Figure 7a. The FTIR spectrum of the CNFs is the same as the one in Figure 1. As indicated by the presence of absorption bands at 811 cm^−1^ and 1232 cm^−1^, and in good agreement with X-ray diffraction, neat PVDF crystallizes in a majority γ-phase, and so do the composite films [15]. Cellulose specific contributions to the FTIR spectra of the composites are weak, and they are primarily observed in the 1000–1110 cm^−1^ range. We did not observe the emergence of any peaks or shoulders indicative of β-PVDF phase formation, regardless of the amount of added cellulose. 

F(γ) was calculated to be 89.5% for the neat PVDF film, and slightly increasing in the composite films to approximately 92% (Figure 7b). Qualitative information regarding α- and γ-phases can be obtained from the FTIR absorbance intensity ratios A_764_/A_874_, and A_1232_/A_874_ where A_764_, A_874_, and A_1232_ are the absorbances at 764, 874 and 1232 cm^−1^, respectively [62]. The calculated ratios, plotted as function of CNF wt %, are shown in Figure 7b. A_1232_/A_874_ gradually increases and plateaus at 2 wt %, indicating a small increase in the γ-phase, whereas the A_764_/A_874_ ratio decreases indicating reduction in the minority α-phase. The FTIR spectra of the TOCNF/PVDF composites, F(γ), and relevant absorbance ratios are shown in Figure 7c,d. F(γ) increases up to 3 wt % and then is somewhat reduced for higher wt %, whereas A_764_/A_874_ and A_1232_/A_874_ follow a trend similar to that of the CNF/PVDF composites. 

When compared to the neutral surface charge of mechanically refined or enzymatically-processed CNFs, the surface charge of TOCNFs with –COONa surface carboxylate groups is negative [63]. Therefore, one would expect that the negatively charged TOCNFs will strongly interact with the positively charged CH_2_ PVDF dipoles. Information regarding specific interactions between dopants and the PVDF matrix can be obtained by examining the FTIR peaks in the 2900–3100 cm^−1^ region. As shown in Figure 8a, the two peaks located at 2981 and 3024 cm^−1^ correspond to the symmetric and asymmetric PVDF CH_2_ stretching bands [64]. The addition of CNFs has a larger effect on the peak position of the asymmetric CH_2_ band, which tends to shift to higher wavenumbers with increasing CNF wt %. Typically, dopants that induce nonpolar to polar phase transitions in PVDF shift both the symmetric and asymmetric CH_2_ bands to lower wavenumbers [65]. Since the CNF/PVDF films remain in the same crystalline phase, the shift in the CH_2_ bands could be indicative of increased disorder in γ-phase chain alignment. The peak position of the symmetric CH_2_ stretching band increases from 2981 to 2982.5 cm^−1^ with increasing TOCNF wt % but decreases to 2980.5 cm^−1^ for the 5 wt % film. On the other hand, the peak position of the asymmetric CH_2_ stretching band increases by approximately 2 cm^−1^ over that of the neat PVDF. Overall, the FTIR absorption bands in the 2900 to 3100 cm^−1^ range are affected more by TOCNFs than CNFs, indicating increased interfacial interaction of the former with the polymer matrix. We hypothesize that this is a consequence of the chemical and morphological changes in the TOCNFs resulting from their oxidation: TOCNFs are morphologically finer than the as-received CNFs, enabling higher surface area per particle and therefore greater availability of charged surface sites for interfacial interactions. Also, as mentioned before, the negative surface charge of TOCNFs is expected to have a stronger interaction with the positive CH_2_ dipoles, as evidenced by the larger shifts of the -CH_2_ FTIR bands. Despite increased interactions between TOCNFs and PVDF, the addition of TOCNFs did not induce a crystalline phase change in the composites.

### 3.3. Thermal Stability and Crystallinity

The first melting curves of the as-prepared PVDF, CNF/PVDF and TOCNF/PVDF composites are shown in Figure 9a,c. Relevant parameters such as the heat of fusion (*ΔH*_f_), peak melting temperature (*T*_m_), and percent crystallinity (*X*_cryst_) are listed in Table 2. Addition of CNFs, as-received or oxidized, shifts *T*_m_ to lower temperatures when compared to that of neat PVDF. Further, *T_m_* of the TOCNF/PVDF composites are generally at a lower temperature than *T_m_* of the CNF/PVDF films. For PVDF, the melting temperatures of the α- and β-phases are between 167–172 °C, whereas the melting temperature of the γ-phase is between 179–180 °C when obtained by crystallization from the melt, and even higher from α- to γ- transformation [15]. The peak temperature of the melting and crystallization peaks also depend on the heating and cooling rates, as well as the molecular weight of PVDF [60]. Further, the melting and crystallization behavior of the composites depends not only on the crystalline phases present in the films, but also on the thickness of the crystalline lamellae and defects. In general, thinner crystalline lamellae melt at lower temperatures. These interrelated factors likely contribute to the reduction of *T_m_* of our composites. 

*X*_cryst_ of the as-prepared CNF/PVDF composites, with the exception of the 3 wt % composite, is higher than that of neat PVDF and also higher than those of as-prepared TOCNF/PVDF composites. On average, CNFs somewhat increase crystallinity in the as-prepared composites, whereas TOCNFs reduce it. The non-isothermal crystallization of the neat film and composites is shown in Figure 9b,d. *T*_cryst_ of the composites are higher than the neat film, with no significant difference between CNFs and TOCNFs. *T*_cryst_ increases with increasing wt % for the CNF/PVDF composites, whereas for the TOCNFs/PVDF composites, it reaches a maximum at 3–4 wt % and slightly decreases for the 5 wt % sample. Similar trends in *T*_cryst_ were observed in nanocellulose/PVDF composites with mixed α-, γ-phases [29], suggesting that CNFs/TOCNFs act as heterogeneous nucleation sites for PDVF crystallization. The melting and crystallization behaviors of the composites suggest an interaction between the polymer matrix and the dopants.

Figure 10 shows the TGA and DTG curves for the CNFs, TOCNFs, and respective PVDF composites, with detailed parameters summarized in Table 2. Generally, the thermal degradation of cellulose involves three steps: dehydration, depolymerization and destruction of glycosidic bonds [66]. CNFs undergo a one-step degradation with *T*_50%_ ~ 337 °C and the maximum rate of mass loss occurring at *T*_p1_ ~ 329 °C. The thermal degradation of TOCNFs is a multistep degradation process, with both *T*_10%_ and *T*_p1_ occurring at approximately ~ 90–100 °C lower than the thermal degradation of CNFs. The lower thermal stability of TOCNFs is attributed to the increase in amorphous content, smaller size and introduction of carboxyl groups on the CNF surface [66,67]

Neat PVDF exhibits one degradation step between 400 and 600 °C with *T*_50%_ ~ 477 °C and *T*_p1_ ~ 470 °C. The thermal decomposition of the 0.5 wt % CNF/PVDF composite is similar to that of the neat PVDF with a smaller DTG peak appearing at a lower temperature (~342 °C) likely corresponding to the thermal degradation of the incorporated CNFs. When the CNF wt % increases beyond 0.5 wt %, the main DTG peak shifts to even lower temperatures (~440 °C). In addition, the main DTG peak in the 400–550 °C range broadens with a high temperature shoulder appearing at 476 °C at 5 wt %. 

The TOCNF/PVDF composites exhibit DTG peaks in the 180-380 °C range, which are correlated to the thermal decomposition of the TOCNFs inside the PVDF matrix. The temperature of the most well defined DTG peak in this temperature range (*T*_p1_) is listed in Table 2. The DTG peaks for all TOCNF/PVDF films occur at a lower temperature than those of the CNF/PVDF films. Similar to the CNF/PVDF films, a higher temperature DTG shoulder also appear, but does not evolve into a distinct peak (as in the case of 5 wt % CNF/PVDF film). As expected, the residual mass of all the composites above 500 °C is higher than that of neat PVDF.

Reports on the thermal stability of CNF/semi-crystalline polymer nanocomposites are limited. However, the effect of CNCs on the thermal stability of polymeric composites has been investigated for CNC/PVDF [31], and CNC/PVDF/poly(methyl methacrylate) (PMMA) composites [59]. In both of these studies, the thermal stability of the composites was enhanced when compared to that of neat films. Since the thermal stability of the CNCs is lower than that of PVDF, these results are somewhat surprising; the authors attribute the increase of thermal stability to the significant increase in the crystallinity of the composites [31,59,62]. For both the CNF/PVDF and TOCNF/PVDF composites, changes in crystallinity over that of neat PVDF are modest; therefore, the reduction in composite thermal stability is dictated by the thermal degradation of the incorporated CNFs or TOCNFs. At approximately 400 °C, the CNFs/TOCNFs have decomposed into char, thereby disrupting the PVDF molecular network and reducing the thermal stability of the composites. 

### 3.4. Mechanical Properties

Representative stress-strain curves from the films are shown in Figure 11. The ultimate tensile strength, tensile modulus, % elongation at break, yield stress and tensile toughness of the films are reported in Table 3. Addition of CNFs or TOCNFs reduces the bulk mechanical properties of the composites over those of neat PVDF. The tensile modulus of the CNF/PVDF composites is comparable to that of neat PVDF at low CNF loadings (less than 2 wt %) and somewhat reduced at higher loadings. The tensile modulus of the CNF/PVDF films is reduced by ~18% when compared to the neat film, and after the reduction, remains relatively constant for up to 5 wt %. The tensile strength of the composites remains approximately constant for all CNF loadings. As evidenced by the reduction in the % elongation at break, the incorporation of CNFs greatly reduces the yielding behavior of the composites. For CNF loading greater than 3 wt %, the % elongation slightly increases, likely due to synergistic effects of fibril entanglement. The observations are somewhat different for the TOCNF/PVDF films. Up to 4 wt %, the tensile strength of the TOCNF/PVDF composites decreases approximately linearly with increasing TOCNF wt % and plateaus to a minimum value of ~ 15 MPa for the 5 wt % sample. The tensile modulus of the TOCNF/PVDF composites shows a maximum for the 1 wt %, with an 18% increase over neat PVDF. The % elongation at break for the TOCNF/PVDF composites significantly decreases, indicating loss of ductility and onset of brittle fracture.

Representative force-displacement curves obtained from the films via nanoindentation are shown in Figure 12a,c for CNF/PVDF and TOCNF/PVDF, respectively. The reduced modulus and hardness are shown in Figure 12b,d and are listed in Table 3. Factors contributing to differences in the values of tensile modulus and modulus extracted from nanoindentation are well known and not will not be discussed here. However, based on our nanoindentation experiments, we can conclude that the surface mechanical properties of the composites are different than those of the bulk, as probed by the tensile tests. For the CNF/PVDF composites, the reduced modulus and hardness remain approximately constant up to 4 wt %. For the 5 wt % composite, a ~52% and ~22% increase in the reduced modulus and hardness is observed. For the TOCNF/PVDF composites, a gradual increase of both reduced modulus and hardness is observed up to ~3 wt %. For higher TOCNF loadings the reduced modulus and hardness are reduced. The reduced modulus and hardness of the 3 wt % TOCNF/PVDF film increase by ~23% and ~25% respectively over neat PVDF. 

CNCs and CNFs are commonly used to reinforce hydrophilic polymers such as poly(ethyleneoxide) (PEO) [13], and poly(vinyl alcohol) (PVA) [68], but dispersion of hydrophilic cellulose in hydrophobic PVDF, is a challenging task. Dispersion of CNFs in hydrophobic PMMA was accomplished by carboxylation of CNFs with COOH surface groups, and removal of water in every step of the fabrication process [14]. In the work of Zhang et al. [31], freeze dried CNCs were first dispersed in DMF, and then added to a PVDF/DMF solution and stirred for 48 h in order to create a homogeneous stable solution. Improvement of mechanical and thermal properties of the composites were attributed to thorough dispersion of the CNCs in the polymer matrix, resulting in a significant increase of crystallinity [31]. Cellulose reinforced composites crosslinked with Diels-Alder adducts were recently demonstrated [69]. In our work, the bulk mechanical properties of our composites obtained from tensile tests indicate that incorporation of CNFs generally reduces bulk tensile properties. For the TOCNF/PVDF composites, the tensile modulus slightly increases at low loadings (1 wt %), but the tensile strength of the composites is linearly reduced with increasing loading. The main reinforcing mechanism of CNFs is their entanglement that assists fibril/matrix and fibril/fibril load transfers [70]. The recovery of the tensile modulus and strength of the CNF/PVDF composites at higher loadings (>2 wt %) could be attributed to increased fiber entanglement and/or increased crystallinity of the composites. The worse bulk performance of the TOCNF/PVDF samples, at higher loadings, is likely due to the loss of fibril crystallinity and transformation to cellulose II following the cellulose oxidation treatment, which reduces the tensile strength of the fibrils [71], as well as the reduced crystallinity of the TOCNF/PVDF films. Interestingly, the surface mechanical properties of the TOCNF/PVDF composites, from 1 to 4 wt %, are better than those of neat PVDF and CNF/PVDF composites of similar loading. 

A schematic of the load-transfer mechanisms during tensile testing of the films is shown in Figure 13. For neat PVDF significant elongation (~38%) and necking occurs prior to fracture. On the contrary, the CNF/PVDF and TOCNF/PVDF composites undergo brittle fracture, with significant reduction in the % elongation at break as the dopant concentration increases. During the initial stages of fracture, stress concentrations at defect sites initiate formation of cracks (see Figure 13b). These defect sites can be voids on the surface and/or interior of the composite films due to vapor-induced phase separation. As the cracks propagate and the polymer matrix fails, CNFs transfer some of the load and their tangles bridge gaps (see Figure 13b). On the contrary, TOCNFs are smaller and more amorphous than CNFs and likely break or pull out easier than CNFs. As the tensile test progresses (Figure 13c), cracks continue to grow and propagate and CNFs/TOCNFs eventually pull out or fail. Inhomogeneous dispersion and fibril agglomeration at higher loadings (>1 wt %) increase defects and decrease fibril/polymer interfacial adhesion, leading to reduction of strength, modulus and % elongation at break.

Our results indicate that the chemical treatment, surface charge and interfacial interactions between CNFs, solvent and the polymer matrix play an important role in the performance of the composites. As-received CNFs have –OH surface groups, which limit their dispersion in most solvents, whereas TOCNFs have negative –COONa surface groups, which should electrostatically repel similarly charged cellulose fibrils and aid dispersion in solvents. However, dispersion of charged particles in solvents also depends on the dielectric constant and viscosity of the solvent; for example, TOCNFs with –COONa surface groups have limited dispersion in DMF but form stable dispersions in DMSO [52,63], which is another suitable solvent for PVDF. Conversely, esterification of as-received CNFs and conversion of TOCNF-COONa into TOCNF-COOH can aid in their dispersion and incorporation of functional polymers such as PVDF.

## 4. Conclusions

We employed a high-shear mixing method to synthesize CNF and oxidized-CNF/PVDF composite films. Oxidized-CNFs have negative surface charges that can strongly interact with positive CH_2_ dipoles and induce crystalline phase changes in the composites. However, for our processing conditions, the composites remain in the same majority crystalline phase, regardless of the amount or type of added cellulose. Our results suggest that certain properties of the composites, such as surface wettability, thermal and mechanical responses, can be altered through incorporation of these dopants. The top surface of the CNF/PVDF composites exhibits increased hydrophobicity due to vapor-induced phase separation. On the contrary, the bottom surface of the TOCNF/PVDF composites exhibits increased hydrophilicity with increasing TOCNF wt %. Further, CNFs somewhat enhance crystallinity in the as-prepared composites, with the 1 wt % film exhibiting ~7% higher crystallinity than that of the neat film. In contrast, TOCNFs decrease crystallinity of the as-prepared composites. Thermal degradation of TOCNF/PVDF composites occurs at lower temperatures than those of the CNF/PVDF composites due to the reduced thermal stability of TOCNFs. In general, addition of CNFs or TOCNFs in PVDF results in reduction of bulk mechanical properties. However, the 1 wt % TOCNF/PVDF film exhibited an 18% increase in the tensile modulus. Surface mechanical properties, probed by nanoindentation, are enhanced under certain loadings. The highest increase in reduced modulus and hardness was observed in the 5 wt % CNF/PVDF sample, and the 3 wt %, 4wt% TOCNF/PVDF composites. Overall, since the ultimate tensile strength of the CNF/PVDF composites is consistently higher than that of the TOCNF/PVDF composites, we believe that CNFs are a better reinforcing dopant than TOCNFs. Our work provides insights on the behavior of PVDF composites loaded with abundant and environmentally friendly, cellulose nanofibrils, mechanically refined or oxidized. Further investigations are warranted to understand the interactions of nanocellulose surface chemistry with the processing methods employed to produce semi-crystalline polymers such as PVDF. 

## Figures and Tables

**Figure 1 polymers-11-01091-f001:**
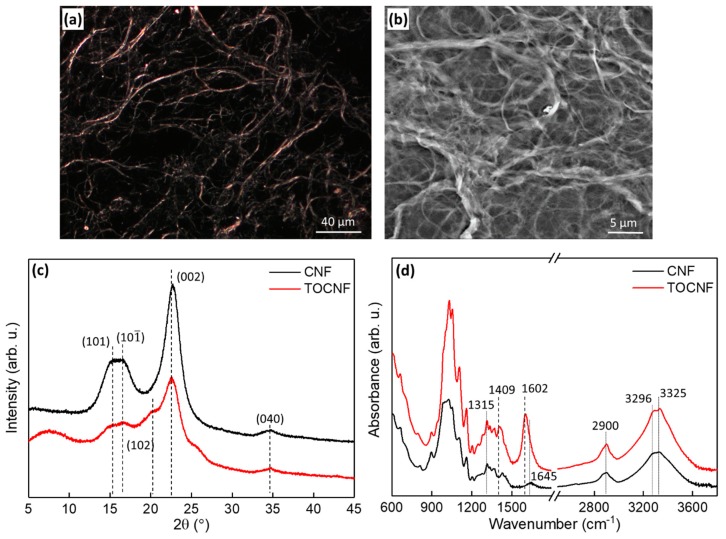
Characterization of the CNFs and TOCNFs. (**a**) Optical images and (**b**) SEM image showing high aspect ratio fibrils from the CNFs. (**c**) X-ray diffraction patterns, and (**d**) FTIR spectra of CNFs and TOCNFs.

**Figure 2 polymers-11-01091-f002:**
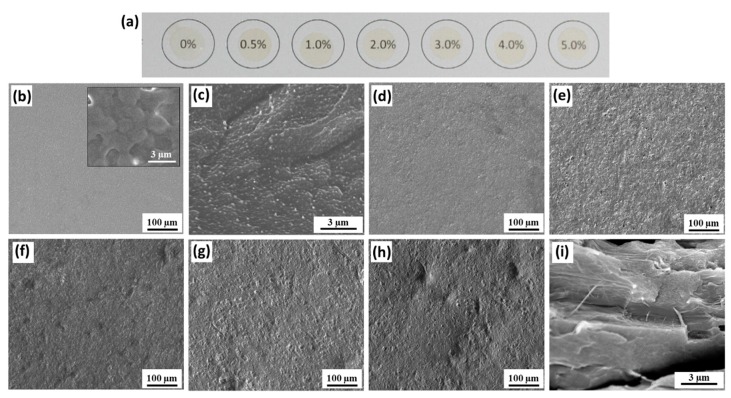
CNF/PVDF film morphology. (**a**) Photograph of the CNF/PVDF composites with increasing CNF wt % (from left to right 0, 0.5, 1, 2, 3, 4 and 5 wt %). (**b**) SEM image of neat PVDF film. Inset shows higher magnification spherulite structure. (**c**) SEM of the cross-section of the freeze-fractured PVDF film. (**d**)–(**h**) SEM images of CNF/PVDF composites with increasing loading: 1, 2, 3, 4 and 5 wt %. (**i**) SEM of the cross-section of the 5 wt % CNF/PVDF freeze-fractured film.

**Figure 3 polymers-11-01091-f003:**
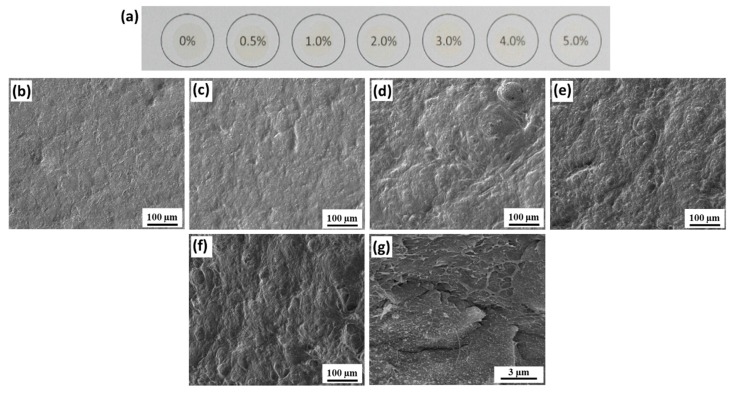
TOCNF/PVDF film morphology. (**a**) Photograph of TOCNF/PVDF composites with increasing TOCNF wt % (from left to right 0, 0.5, 1, 2, 3, 4 and 5 wt %). (**b**)–(**f**) SEM images of TOCNF/PVDF composites with increasing loading: 1, 2, 3, 4 and 5 wt %. (**g**) SEM of the of the cross-section of the 5 wt % TOCNF/PVDF freeze-fractured film.

**Figure 4 polymers-11-01091-f004:**
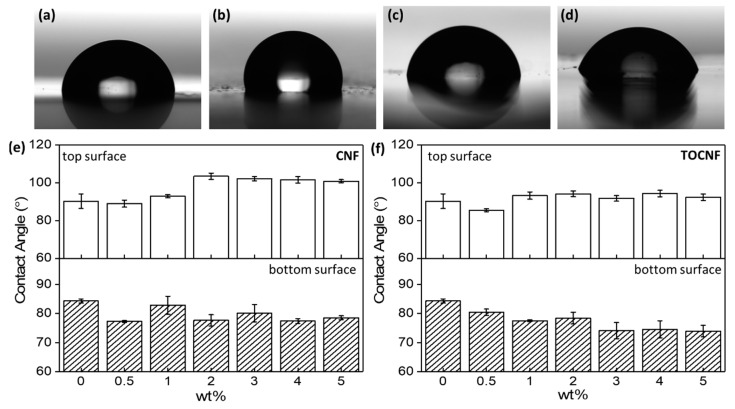
Contact angle measurement for (**a**) neat PVDF top surface, (**b**) 2 wt % CNF/PVDF top surface, (**c**) neat PVDF bottom surface and (**d**) 5 wt % TOCNF/PVDF. Comparison of water contact angle of (**e**) top and (**f**) bottom (constrained) surface of PVDF, CNF/PVDF and TOCNF/PVDF composites.

**Figure 5 polymers-11-01091-f005:**
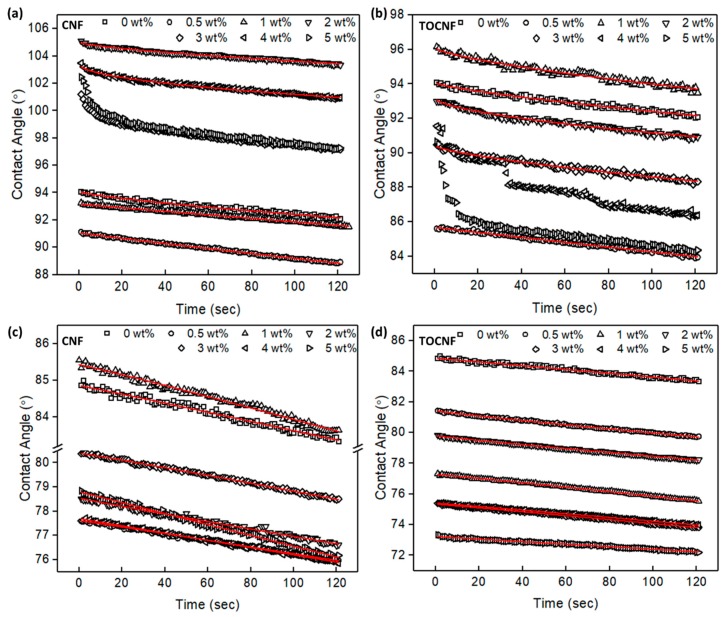
Time-dependent contact angle of (**a**), (**b**) top (free), and (**c**), (**d**) bottom (constrained) surface of the CNF/PVDF and TOCNF/PVDF films. Experimental data are shown as symbols and fits as solid lines.

**Figure 6 polymers-11-01091-f006:**
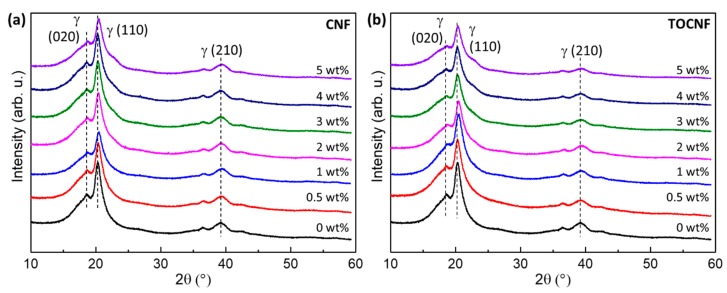
X-ray diffraction patterns for (**a**) CNF/PVDF and (**b**) TOCNF/PVDF films as function of cellulose wt %.

**Figure 7 polymers-11-01091-f007:**
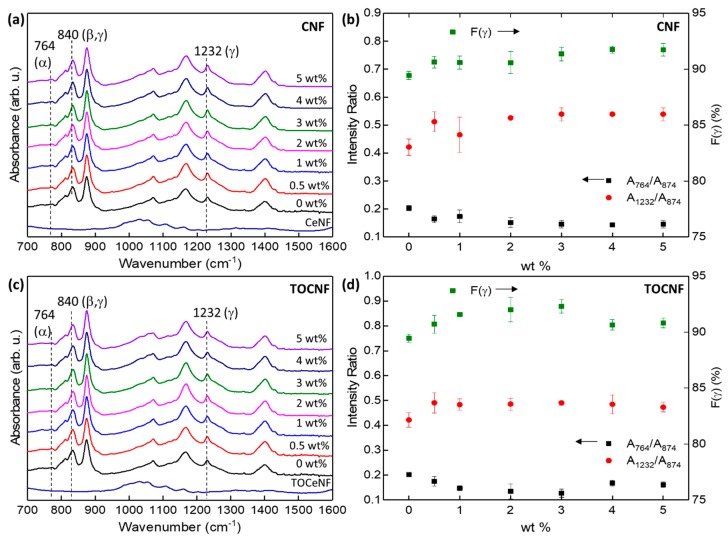
FTIR spectra of (**a**) CNF and CNF/PVDF composites, (**b**) calculated A_765_/A_874_, A_1232_/A_874_ ratios and F(γ) for CNF/PVDF composites. (**c**) FTIR spectra of TOCNF and TONCF/PVDF composites, (**d**) calculated A_765_/A_874_, A_1232_/A_874_ ratios and F(γ) for TOCNF/PVDF composites.

**Figure 8 polymers-11-01091-f008:**
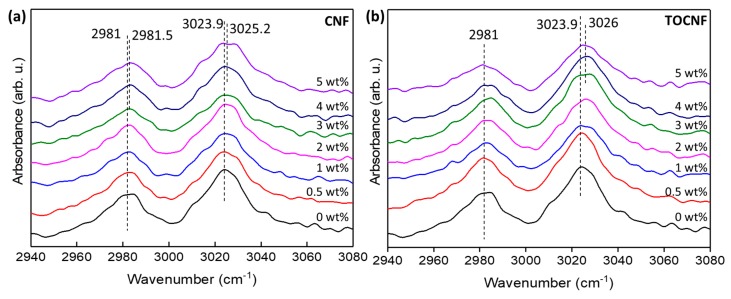
FTIR spectra of symmetric and asymmetric -CH_2_ vibrations for (a) CNF/PVDF, and (b) TOCNF/PVDF composites.

**Figure 9 polymers-11-01091-f009:**
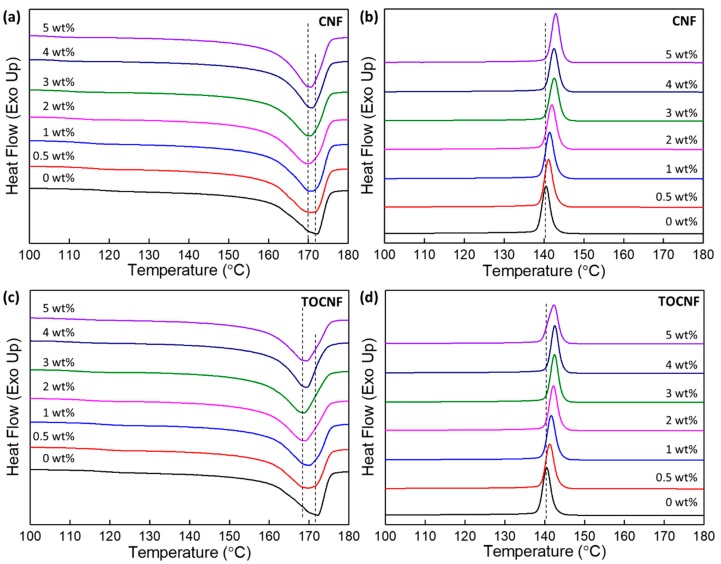
Melting and crystallization behaviors of the films measured by DSC. (**a**) First melting curves and (**b**) crystallization curves of CNF/PVDF composites. (**c**) First melting curves and (**d**) crystallization curves of TOCNF/PVDF composites.

**Figure 10 polymers-11-01091-f010:**
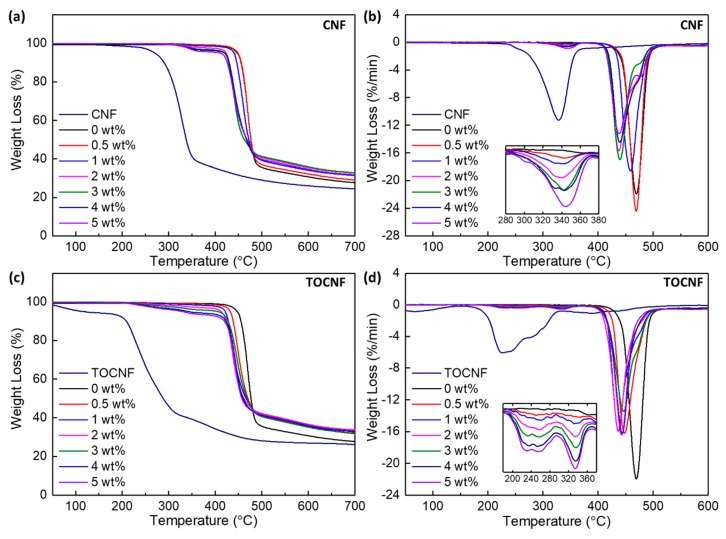
TG and DTG curves of CNF, TOCNF, CNF/PVDF and TOCNF/PVDF composites.

**Figure 11 polymers-11-01091-f011:**
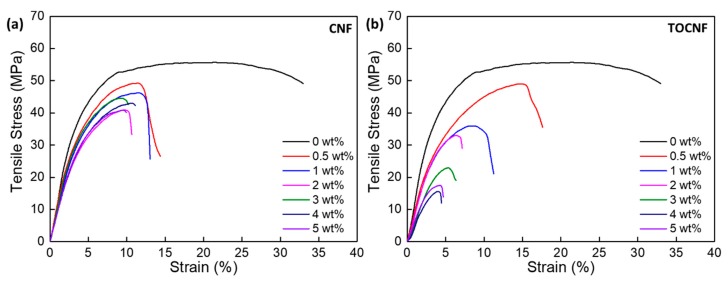
Representative tensile stress-strain curves for (**a**) CNF/PVDF, and (**b**) TOCNF/PVDF composites.

**Figure 12 polymers-11-01091-f012:**
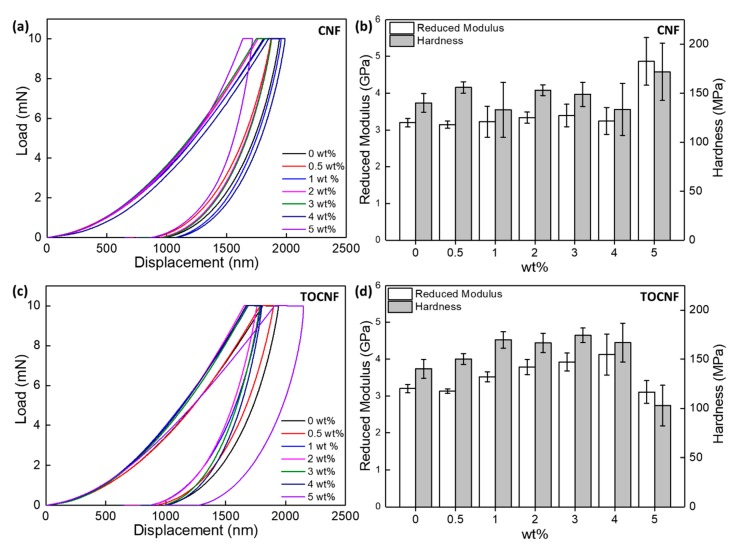
Nanoindentation load-displacement curves from (**a**) CNF/PVDF and (**c**) TOCNF/PVDF composites and corresponding reduced modulus and hardness (**b**), (**d**).

**Figure 13 polymers-11-01091-f013:**
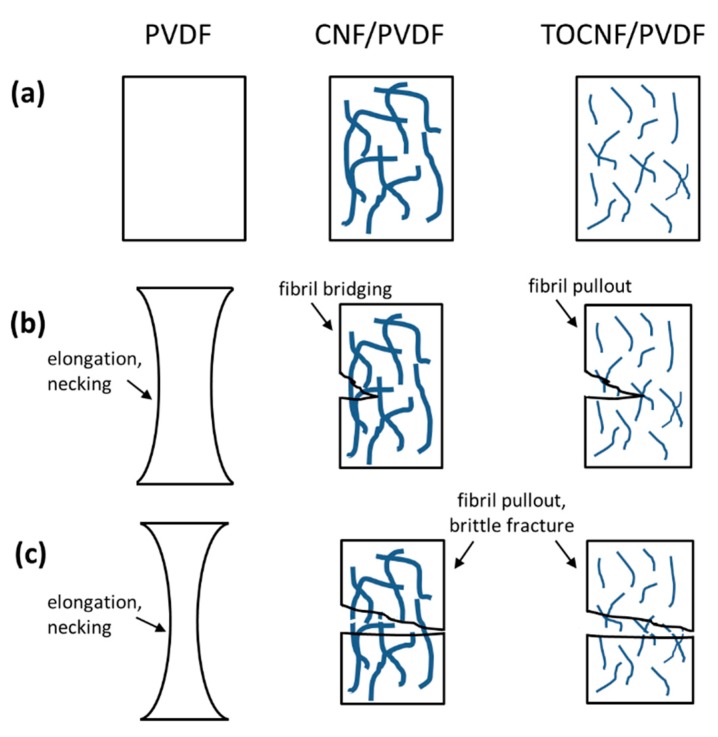
Schematic of load-transfer mechanisms during tensile testing of PVDF, CNF/PVDF and TOCNF/PVDF composites. (**a**), (**b**), (**c**) indicate the various stages of film evolution during the progression of the tensile test.

**Table 1 polymers-11-01091-t001:** Summary of Equation (1) fitting results for PVDF, CNF/PVDF, and TOCNF/PVDF films.

	top surface			bottom surface		
wt %	*θ* _i_	*k* (×10^−4^ s^−1^)	*n*	*θ* _i_	*k* (×10^−4^ s^−1^)	*n*
0	94.08	5.6	0.75	84.87	1.4	1.00
CNF						
0.5	91.07	2.6	0.95	77.63	1.8	1.00
1	93.16	1.3	1.00	85.48	1.8	1.00
2	105.20	5.3	0.70	78.54	2.4	0.97
3	-	-	-	80.43	2.0	1.00
4	103.42	2.1	0.51	77.70	1.9	1.00
5	-	-	-	78.80	2.8	1.00
TOCNF						
0.5	85.68	1.9	0.98	81.43	2.9	0.89
1	96.10	8.9	0.70	77.32	1.8	1.00
2	93.20	14.9	0.58	79.80	2.5	0.91
3	90.43	7.8	0.71	75.36	1.7	1.00
4	-	-	-	75.43	1.6	1.00
5	-	-	-	73.23	1.2	0.99

**Table 2 polymers-11-01091-t002:** DSC and TGA results from CNF, TOCNF, PVDF, CNF/PVDF, and TOCNF/PVDF composites.

wt%	*T_m_* (°C)	*T_cryst_* (°C)	*Δ**H_f_* (J/g)	*X_c_* (%)	*T_10%_* (°C)	*T_50%_* (°C)	*T_p1_* (°C)	*T_p2_* (°C)	*T_p3_* (°C)
0	172.2	140.6	52.1	49.8	455.0	476.6	-	469.5	-
CNF	-	-	-	-	289.9	337.5	328.5	-	-
0.5	170.6	141.1	54.7	52.6	455.7	476.1	-	467.8	-
1	170.8	141.4	55.0	53.1	446	472.1	341.2	456.8	-
2	170.1	142.0	52.8	51.5	429.7	467.5	341.4	436.1	-
3	170.4	142.6	48.2	47.5	428.5	462.4	341	443.2	-
4	170.9	142.5	50.4	50.2	429.5	466.9	338.5	439.4	-
5	170.5	142.9	54.3	54.6	425.5	468.5	341.2	435.1	476.1
TOCNF	-	-	-	-	209.1	288	224.1	-	-
0.5	169.8	141.3	51.0	49.0	438.2	470.1	-	449.7	-
1	169.9	141.6	50.7	49.0	427.9	466.2	331.8	436.5	-
2	168.9	142.2	48.6	47.4	423.7	461.6	332.9	437.3	-
3	168.6	142.5	49.5	48.8	427.1	468.3	333	446.6	-
4	169.4	142.5	46.2	46.0	422.9	459.7	332.8	444.6	-
5	169.4	142.3	49.0	49.0	419.6	457	333.4	442.7	-

**Table 3 polymers-11-01091-t003:** Mechanical properties of PVDF, CNF/PVDF and TOCNF/PVDF composites obtained from tensile and nanoindentation experiments.

wt%	Tensile Test					Nanoindentation	
	Ultimate Tensile Strength (MPa)	Tensile Modulus (GPa)	Elong. at Break (%)	Yield Stress (MPa)	Tensile Toughness (10^3^ kJ/m^3^)	Reduced Modulus (GPa)	Hardness (MPa)
0	54.7 ± 5.9	1.12 ± 0.03	37.7 ± 4.2	29.2 ± 1.4	17.6 ± 3.1	3.20 ± 0.12	140.0 ± 9.5
CNF							
0.5	43.4 ± 6.8	1.22 ± 0.18	14.1 ± 1.1	21.1 ± 3.2	4.5 ± 1.1	3.15 ± 0.10	155.8 ± 5.8
1	43.5 ± 9.5	1.08 ± 0.28	11.7 ± 2.9	20.4 ± 1.7	3.6 ± 1.5	3.22 ± 0.42	133.0 ± 28
2	41.1 ± 0.5	0.75 ± 0.04	19.0 ± 7.6	21.7 ± 1.9	6.5 ± 3.0	3.33 ± 0.15	153.1 ± 5.5
3	45.6 ± 1.4	0.94 ± 0.04	10.2 ± 1.0	21.6 ± 1.2	3.4 ± 0.1	3.40 ± 0.31	148.9 ± 12.3
4	47.1 ± 3.9	1.09 ± 0.01	10.8 ± 0.3	25.1 ± 0.4	3.7 ± 0.3	3.25 ± 0.13	133.5 ± 26.5
5	42.5 ± 1.6	1.05 ± 0.08	11.7 ± 2.1	21.5 ± 4.8	3.4 ± 0.7	4.86 ± 0.65	171.7 ± 29.2
TOCNF							
0.5	47.8 ± 5.4	1.13 ± 0.06	18.1 ± 5.7	21.1 ± 3.1	6.6 ± 2.5	3.13 ± 0.06	150.1 ± 5.6
1	41.5 ± 4.9	1.33 ± 0.05	12.2 ± 0.9	23.8 ± 2.6	3.8 ± 0.7	3.52 ± 0.14	169.6 ± 8.5
2	33.3 ± 0.6	0.75 ± 0.06	8.1 ± 0.3	20.7 ± 0.6	1.8 ± 0.1	3.78 ± 0.20	166.5 ± 9.9
3	24.6 ± 3.8	0.65 ± 0.12	6.6 ± 0.6	17.9 ± 1.2	1.0 ± 0.2	3.92 ± 0.24	174.3 ± 7.4
4	15.8 ± 1.8	0.20 ± 0.02	5.1 ± 0.6	11.4 ± 1.1	0.4 ± 0.1	4.13 ± 0.56	166.9 ± 19.8
5	14.6 ± 2.7	0.48 ± 0.04	4.6 ± 0.9	11.4 ± 0.7	0.3 ± 0.2	3.11 ± 0.31	102.8 ± 20.6

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
