# Peer review of "Effect of Cellulose Nanofibrils and TEMPO-mediated Oxidized Cellulose Nanofibrils on the Physical and Mechanical Properties of Poly(vinylidene fluoride)/Cellulose Nanofibril Composites"

_polymers, 2019, doi:10.3390/polym11071091_

Round 1
Reviewer 1 Report
The paper cannot be published in the present form. My suggestions were not fully considered in the revised version of the paper, In particular, the effective frame rate (1sec) of the experimental contact angle experiments should allow to perform a fitting analysis of the water contact angle vs time trends. This analysis is necessary for a correct and deeper investigation of the surface properties of the composite materials.
Author Response
Point 1: The paper cannot be published in the present form. My suggestions were not fully considered in the revised version of the paper, In particular, the effective frame rate (1sec) of the experimental contact angle experiments should allow to perform a fitting analysis of the water contact angle vs time trends. This analysis is necessary for a correct and deeper investigation of the surface properties of the composite materials.
Response 1: Per reviewer request we have carried out time-dependent contact angle experiments (1 sec frame rate, 120 seconds duration) on all the films, on both the top and bottom surfaces for each film. We have revised the appropriate manuscript sections and have discussed the new experimental data and fitting results (lines 306-333). We added a new figure showing the time-dependent contact angle measurements (Figure 5), and new table (Table 1) summarizing fitting results from the experimental data. The fittings results are presented in more detail in Table S2 of the Supporting Information.
Reviewer 2 Report
In this manuscript, the effect of cellulose nanofibrils (CNFs) and TEMPO-mediated oxidized CNFs on physical and mechanical properties of poly (vinylidene fluoride)/CNF composites is investigated. This manuscript is well characterized in detail and discussed. However, when I gone through the manuscript, I found some critical issues as follows:
1. Why authors used carboxylated-CNFs in addition to unmodified-CNFs?. Is it beneficial over CNFs. A reasonable description should be given in the Results and Discussion section and conclusion section.
2. Authors have performed wettability test for top and bottom surface of PVDF, but for CNF/PVDF composite film only top surface is analysed, while for TOCNFs/PVDF composite films only bottom surface is analysed and compared to each other. I am surprised how authors have concluded these difference surface organization of molecules or microstructures of two different composite films?.
3. Authors should provide the schematic model of the load-transfer mechanism inside the composite system, because mechanical results are little surprising, in my opinion, compared to the composite of hydrophilic polymer and cellulose nanofibrils. However, Negative or positive results always give a better insight in fabricating value added products and must be discussed with reasoning. In my opinion, the incorporation of CNCs in polymeric matrix shows increased modulus and decreased elongation at break and follows the order with increased content of CNCs. In case of CNFs, these show high load-transfer mechanism with improved modulus and elongation at break. For this reason, authors may referred following articles: PVDF/CNFs (International Journal of Engineering &Technology, 7 (3.12) (2018) 1025 -1029)
4. With the incorporation of CNFs and TEMPO-mediated oxidized-CNFs, the thermal stability is decreased. In my opinion, this is due to the disruption of the molecular network of PVDF. However, authors should describe how CNFs and TOCNFs behave within the PVDF structure-network for describing thermal analysis. For thermal analysis, the following article may be referred for needed description: Characterization of cellulose nanocrystals produced by acid-hydrolysis from sugarcane bagasse as agro-waste. Journal of Materials Physics and Chemistry, 2(1), 2014, 1-8 and
5. In conclusion section, which nanoreinforcement is better should be reported.
After this minor revision, this manuscript can be considered for publication in this journal.
Author Response
Point 1: Why authors used carboxylated-CNFs in addition to unmodified-CNFs?. Is it beneficial over CNFs. A reasonable description should be given in the Results and Discussion section and conclusion section.
Response 1: Carboxylation of the CNFs creates negative surface changes that can theoretically interact with positively charged CH2 PVDF dipoles resulting in crystalline phase changes in PVDF. For example, negatively charged nanoparticles have been reported to induce phase transformation from either non-polar to polar or from gamma phase polar to beta phase polar PVDF (Tamang et al. [58], Martins et al. [61]). A surprising result of our study is that addition of negatively changed TOCNFs did not induce a crystalline phase change in the TOCNF/PVDF composites (despite our similar processing conditions with reference [58] which all remain in the gamma polar phase, regardless of the amount of added TOCNFs.
We have revised the Introduction (lines 98-99), the Results and Discussion section (lines 235-237, 342) as well as the Conclusion (lines 571-572) including the above justification for using TOCNFs in addition to CNFs in the composites.
Point 2: Authors have performed wettability test for top and bottom surface of PVDF, but for CNF/PVDF composite film only top surface is analysed, while for TOCNFs/PVDF composite films only bottom surface is analysed and compared to each other. I am surprised how authors have concluded these difference surface organization of molecules or microstructures of two different composite films?
Response 2: We carried out wettability studies on both the top and bottom surfaces of all films including CNF/PVDF and TOCNF/PVDF; however in the Conclusion we only discussed our observations for the top surface of CNF/PVDF and bottom surface of TOCNF/PVDF. All the results (top and bottom surfaces) were presented in Figure 4, as well as in (new) Figure 5. We have revised our manuscript with expanded discussion of the contact angle results for all composites films and corresponding surfaces (top surface of PVDF, CNF/PVDF, TOCNF/PVDF, bottom surface of PVDF, CNF/PVDF, TOCNF/PVDF). (lines 285-295)
Point 3: Authors should provide the schematic model of the load-transfer mechanism inside the composite system, because mechanical results are little surprising, in my opinion, compared to the composite of hydrophilic polymer and cellulose nanofibrils. However, Negative or positive results always give a better insight in fabricating value added products and must be discussed with reasoning. In my opinion, the incorporation of CNCs in polymeric matrix shows increased modulus and decreased elongation at break and follows the order with increased content of CNCs. In case of CNFs, these show high load-transfer mechanism with improved modulus and elongation at break. For this reason, authors may referred following articles: PVDF/CNFs (International Journal of Engineering &Technology, 7 (3.12) (2018) 1025 -1029)
Response 3: We agree with the reviewer that the morphology of dopants affects the mechanical properties of composite materials. For example, carbon nanotube polyamide 12 composite films exhibit improved modulus and elongation-at-break when compared to composites made with graphene nanoparticles (for example Chatterjee et al., Nanotechnology 22 (2011) 275714). Also the elongation-at-break for all composites was lower than that of neat polyamide 12. When comparing carbon nanotube and graphene electrospun polyamide 12 composite fibers, the graphene composites showed higher elastic modulus and elongation-at-break (Chatterjee et al., Chemical Physics Letters 557 (2013) 92-96).
Comparative studies of CNC versus CNF reinforcement effects in polymer composites are limited. In hydrophilic polyethylene oxide (PEO), for the same dopant loading, CNF/PEO composite films induced higher strength and modulus than CNC/PEO composites, but lower strain-at-failure likely because of agglomeration (Xu et al. 2013 [13]). For both types of dopants, higher loadings lead to elongation-at-break smaller than that of neat PEO. On the contrary, CNC/PEO nanofiber mats had higher modulus and tensile strength than CNF/PEO nanofiber mats; both types of nanofiber mats exhibited similar strain-at-failure, which was higher than that of neat PEO (Xu et al. Macromolecules 2014, 47, 3409-3416). Based on these examples, we believe it is difficult to predict how dopants will affect the tensile properties of composites because experimental results are often contradictory. Load transfer mechanisms in the composites are the result of a sometimes complex interplay between dopant morphology and dopant-matrix interfacial interactions.
In response to the reviewer comment, we have added a new figure (Figure 13) illustrating potential load-transfer mechanisms in PVDF, CNF/PVDF and TOCNF/PVDF composites. We have also revised our manuscript with a short discussion of Figure 13. (lines 542-558)
Point 4: With the incorporation of CNFs and TEMPO-mediated oxidized-CNFs, the thermal stability is decreased. In my opinion, this is due to the disruption of the molecular network of PVDF. However, authors should describe how CNFs and TOCNFs behave within the PVDF structure-network for describing thermal analysis. For thermal analysis, the following article may be referred for needed description: Characterization of cellulose nanocrystals produced by acid-hydrolysis from sugarcane bagasse as agro-waste. Journal of Materials Physics and Chemistry, 2(1), 2014, 1-8 and
Response 4: We have revised the discussion regarding the thermal stability of the composites with discussion on how the thermal decomposition of the CNFs/TOCNFs inside the PVDF matrix disrupt the PVDF structure thereby reducing its thermal stability (lines 442-444, 447, 448, 452, 456, 469-473). We have added the listed publication as a new reference.
Point 5: In conclusion section, which nanoreinforcement is better should be reported.
Response 5: We have revised the conclusion as follows: “Overall, since the ultimate tensile strength of the CNF composites is consistently higher than that of the TOCNFs composites, we believe that CNFs are better reinforcing dopants than TOCNFs.” (lines 587-589)
Round 2
Reviewer 1 Report
The paper was improved according to the reviewers' comments. I recommend its publication in the current form.
This manuscript is a resubmission of an earlier submission. The following is a list of the peer review reports and author responses from that submission.
Round 1
Reviewer 1 Report
This manuscript deals with the development of cellulose nanofibril-poly(vinylidene fluoride) composites. The writing of this manuscript is clear and the methods used are presented well.
This manuscript has a number of weaknesses. This work is not significant and has a low level of novelty. The authors aim at using bio-based reinforcement for poly(vinylidene fluoride) (PVDF), whereas the state of the art for PVDF-based composites and their properties is not presented at all in the introduction section. The use of a given type of nanocellulose, CNFs, as a reinfocement is not justified. Mixing of hydrophilic reinforcement in a hydrophobic polymer matrix is challenging and should be tackled to achieve the mechanical reinforcement. In this study, however, the authors do not tackle this issue and destroy the initial strength of PVDF by adding nanocellulose and demonstrate the reduction of the mechanical properties. Therefore, the results do not demonstrate any development and are not interresting for the reader. It is not clear how TOCeNF sample was obtained, by post oxidation of CeNF or oxidation of pulp and further fibrillation. Therefore, the work becomes not reproducible.
Some other errors and concerns in the manuscript:
Line 15 and further: Cellulose nanofibrils are usually abbreviated as CNFs, not CeNFs. CNFs is suggested to be used.
Line 15: The Abstract starts with a sentence of a state of the art. This can be appropriate if this sentence presents a problem that is further solved in the paper. However, the first sentence does not contain any problem and is not logically connected to the subsequent text.
Line 16: It is not clear which hierarchical structure of CNFs is referred to. Macroscopic fibers have pronounced hierarchical structure, whereas CNFs consist of ordered and disordered regions along the fiber axis. The strength originates from highly aligned molecules.
Line 17: TEMPO is not an oxidant. Therefore, it is correct to call the process as TEMPO-mediated oxidation.
Line 18: This sentence is not logically correct. How do you “use CNFs as reinforcing agent in PVDF using high-shear mixing”?
Line 21: it is not clear what do the authors mean under promote or suppress crystallinity.
Line 25-27: the sentence is not clear and should be rephrased. It is written that at low loadings the tensile modulus remains comparable, which is followed by presenting an 18% increase at 1% CNFs, which for me is low loadings.
Line 40: Reinforcements are normally added at low solids content in petroleum-based composites. Therefore, it is not clear why these fillers “manufactured with non-renewable material … put a significant strain on the environment”. The main constrain is put by non-sustainable matrix, not a filler.
Line 43: please give some examples of such hazardous fillers.
Line 46: cellulose does not consist of linear chains of glucose molecules. It consists of anhydroglucose units.
Line 47-49: molecules do not assemble into rod-shaped particles known as cellulose nanocrystals. These molecules assemble into nanofibrils, where one molecule can pass through several amorphous and crystalline domains. However, researchers developed methods how to extract single crystalline regions.
Line 50: If cellulose nanocrystals were ideal reinforcing nanofillers, as is written here, they would be already used in various commercial applications. However, it is not known about any of such mass production applications so far.
Line 52: I hear for the first time about this “fibril interlocking mechanisms”. Please explain more what is that.
Line 70: speculations like “CeNCs can be more expensive” should be supported by some data. As of today, CNCs can be produced and supplied as dry powders, whereas CNFs only as aqueous suspensions at ca. 2 wt.%. Therefore the cost of using CNC in the production chain may be much lower compared to that of CNFs.
Line 79: the message given in the state of the art is that there is a lot of non-sustainable fillers, which should be replaced by bio-based ones, like nanocellulose. What is the state of the art of PVDF composites? Which fillers are commonly used and which mechanical properties are achieved?
Line 81: it is not clear how by dispersing CNFs in PVDF solutions films are obtained? Some drying step is obviously missing.
Line 89-93: what do you mean under “dry DMF”? DMF is a liquid.
Line 96: what is the source of “as-received cellulose”?
Line 100: In order to produce nanofibrils from cellulosic pulp that was oxidized with TEMPO catalyst, a fibrillation step should be applied, like homogenization.
Line 113: Which solvent was used to dissolve CNF?
Line 180: the Segal method should go into Materials and Methods section.
Figure 2c: “SEM of freeze-fractured PVDF film.” Does it refer to the cross-section?
Line 366: From the graphs it is clear that the crystallization temperature increases both for CNF and TOCNF. It is not clear why the authors consider that CNF promote crystallization, whereas TOCNF suppress crystallization?
Line 417: It should be highlighted that AFTER THE REDUCTION it remains relatively constant for up to 5 wt%.
Line 443: “incorporation of CeNFs does not significantly improve tensile properties”. The presented data shows that incorporation of CeNFs REDUCES tensile properties.
Reviewer 2 Report
This manuscript studied the Effect of Mechanically Refined and Oxidized Cellulose Nanofibrils on the Physical and Mechanical Properties of Poly(vinylidene fluoride)/Cellulose Nanofibril Composites. The authors used cellulose nanofibrils to reinforce PVDF. However, almost all the properties of the cellulose nanofibril/PVDF composites became worse compared with the neat PVDF. Especially, the tensile strengths of the composites were much lower compared with both of neat cellulose nanofibril films and PVDF films. As the authors mentioned, the reason may be the bad dispersion of cellulsoe nanofibril in DMF and also in PVDF matrix. The authors should improve the method to preapre the composites to improve their properties.
Reviewer 3 Report
The authors investigated the effect of mechanically refined and oxidized cellulose nanofibrils on the physical and mechanical properties of PVDF/NC composite films. The results obtained are quite interesting. The manuscript is well written and organized. I think this paper can be accepted with minor revision.
Equations (1), (2), and (3) is better to be moved to experimental section.
Reviewer 4 Report
The paper is focused on the preparation and characterization of nanocomposite films based on poly(vinylidene fluoride) and cellulose nanofibrils. The topic is interesting for readers of Polymers. The presentation and discussion of the experimental results could be partly improved. Based on these considerations, I recommend the publication of the submitted MS after the following revisions:
-Some examples of stress vs strain curves could be presented.
-Is it possible to determine the yelding point from the analysis of stress vs strain curves.
-I suggest to determine the stored energy up to the film breaking by integrating the stress vs. strain curves.
-Did the authors monitor the time variation of water contact angle? As reported in literature for biocomposite films with variable structure (such as multilayer chitosan/halloysite [New J. Chem., 2018,42, 8384-8390] and uniform pectin/coffee grounds [Carb. Polym. 2017, 170, 198–205]), the analysis of the water contact angle vs time trends provide the kinetic constant and the exponential parameter, which is related to the absorption/spreading contributions. Both parameters can be very useful to investigate the effect of the multilayer morphology on the wettability properties of the prepared materials.
-Introduction could be updated by quoting recnt articles on the preparation of nanocomposite films based on cellulose matrix (Nanomaterials 2017, 7, 199), fibers (Polymers 2019, 11(1), 117) and nanofibrils (Polymers 2019, 11(1), 153).